# Analysis of Whole-Genome Sequences of Pathogenic Gram-Positive and Gram-Negative Isolates from the Same Hospital Environment to Investigate Common Evolutionary Trends Associated with Horizontal Gene Exchange, Mutations and DNA Methylation Patterning

**DOI:** 10.3390/microorganisms11020323

**Published:** 2023-01-27

**Authors:** Ilya S. Korotetskiy, Sergey V. Shilov, Tatyana Kuznetsova, Bahkytzhan Kerimzhanova, Nadezhda Korotetskaya, Lyudmila Ivanova, Natalya Zubenko, Raikhan Parenova, Oleg N. Reva

**Affiliations:** 1Scientific Center for Anti-Infectious Drugs, Almaty 050060, Kazakhstan; 2Centre for Bioinformatics and Computational Biology, Department of Biochemistry, Genetics, and Microbiology, University of Pretoria, Pretoria 0002, South Africa

**Keywords:** hospital infection, bacterial pathogen, PacBio sequencing, genotyping, MLST, virulence factor, methylomics

## Abstract

Hospital-acquired infections are a generally recognized problem for healthcare professionals. Clinical variants of Gram-negative and Gram-positive pathogens are characterized with enhanced antibiotic resistance and virulence due to mutations and the horizontal acquisition of respective genetic determinants. In this study, two *Escherichia coli*, two *Klebsiella pneumoniae*, three *Pseudomonas aeruginosa*, two *Staphylococcus aureus*, one *Staphylococcus epidermidis* and one *Streptococcus pneumoniae* showing broad spectra of antibiotic resistance were isolated from patients suffering from nosocomial infections in a local hospital in Almaty, Kazakhstan. The aim of the study was to compare general and species-specific pathways of the development of virulence and antibiotic resistance through opportunistic pathogens causing hospital-acquired infections. The whole-genome PacBio sequencing of the isolates allowed for the genotyping and identification of antibiotic resistance and virulence genetic determinants located in the chromosomes, plasmids and genomic islands. It was concluded that long-read sequencing is a useful tool for monitoring the epidemiological situation in hospitals. Marker antibiotic resistance mutations common for different microorganisms were identified, which were acquired due to antibiotic-selective pressure in the same clinical environment. The genotyping and identification of strain-specific DNA methylation motifs were found to be promising in estimating the risks associated with hospital infection outbreaks and monitoring the distribution and evolution of nosocomial pathogens.

## 1. Introduction

Despite the introduction of effective antibiotic treatment regiments, hospital-acquired infections continue to cause morbidity and mortality around the world. Hospital-acquired infections occur in 7–10% of hospitalized patients [1]. The most common diseases are pneumonia and bloodstream, urologic and postoperative infections [2,3,4]. The main causes of acute nosocomial infections are medical procedures, bad adherence of patients to prescribed antibiotics, inadequate hospital management and patient health status [5]. Common sources of infection include germ carriage by medical staff or patients and surface contamination [6]. The hospital environment provokes the selection and evolution of bacterial clonal lines with increased virulence and multidrug resistance [7,8]. Antibiotic-resistant isolates are frequent among *Escherichia coli*, *Klebsiella pneumonia*, *Staphylococcus aureus* and *Streptococcus pneumoniae*. The WHO (https://www.who.int, accessed on 6 January 2023, and www.who.int/news/item/06-05-2022-who-launches-first-ever-global-report-on-infection-prevention-and-control, accessed on 6 January 2023) considers enterobacteria resistant to vancomycin and carbapenem, as well as methicillin-resistant *Staphylococcus,* as major threats to global health systems [9]. A comprehensive overview of Gram-negative agents of hospital infections was given by Peleg and Hooper [10]. Antibiotic resistance may be associated with the acquisition of genes encoding either alternative variants of the molecular targets of antibiotics or enzymes, which decompose antibiotic molecules or remove them from cells. The mutation of target macromolecules, including proteins and functional RNAs, is another mechanism of acquired drug resistance [11]. The research question of this study is which common or specific mechanisms of acquired antibiotics resistance could be exploited by different opportunistic pathogens inhabiting the same clinical environment. Additionally, a hypothesis is checked as to whether the risk of the emergence of hospital infections depends on epigenetic modifications of the chromosomal DNA of pathogens.

Due to limited choice in effective drugs, nosocomial infections cause severe therapeutic problems by threatening public health, increasing the duration of hospital stay and the cost of patient treatments [12]. Timely infection control and prevention are necessary to tackle the problem and reduce the risk of hospital infection outbreaks. Laboratory diagnostics should be improved, especially in terms of the determination of resistance to antibacterial drugs and disinfectants, and the detection of newly appeared pathogens able to cause hospital infections. Accurate and fast genotyping methods are particularly important for obtaining follow-up information, including the origin of an infection, evolutionary pathways of pathogens characterized by an enhanced virulence and the monitoring of general epidemiological situations. Multilocus sequence typing (MLST) allows for the identification of individual sequence types (STs) of microorganisms, which correspond to specific lineages and clonal complexes. Different STs of pathogenic bacteria are often geographically isolated and characterized with distinguishable phenotypes, different propensities to cause disease outbreaks and the severity of disease manifestations. This is why MLST and serotyping are popular techniques aiding in the estimation of the epidemiological risks of discovered pathogenic bacterial isolates. For instance, outbreaks of intrahospital enterobacterial infections in the Eurasian region are most often associated with *Escherichia coli* ST43 and ST131, whereas several other STs of *E. coli* cause hospital infection in other geographical regions around the globe [13]. Hospital-acquired infections caused by various pathogens are frequent in Central Asian hospitals, particularly in Kazakhstan. These infections are generally associated with surgical sites, followed by ventilator-associated pneumonia, catheter-related bloodstream infections and catheter-associated urinary tract infections [14]. The most common causative agents of nosocomial infections in Kazakhstan are *Pseudomonas aeruginosa*, *Escherichia coli*, *Klebsiella pneumoniae*, *Acinetobacter baumannii*, *Enterococcus faecalis*, *Staphylococcus epidermidis* and *S. aureus* (MRSA and non-MRSA) [15,16]. Many of these isolates show broad spectra of resistance to methicillin, ceftriaxone, cefotaxime, cefuroxime and some other antibiotics; however, whole-genome sequencing to determine specific genetic determinants of antibiotic resistance has not been performed in these studies.

Numerous investigations were performed around the world to identify the virulence factors and determine the mechanisms of their acquisition through opportunistic pathogens causing sudden outbreaks of hospital infections. Based on these outputs, several useful databases of genetic virulence and antibiotic resistance determinants have been created [17,18,19]. However, the strategies of the majority of these studies were to use collections of bacterial strains belonging to the same species of a taxonomic group of pathogens isolated in an enclosed geographic region, whereas the coevolutionary events in taxonomically distant pathogens inhabiting the same environment are generally understudied. In this study, a collection of Gram-negative and Gram-positive pathogenic bacteria isolated from Syzganov’s National Scientific Center of Surgery in Almaty (Kazakhstan), characterized with broad spectra of antibiotic resistance, was selected for PacBio sequencing followed by genotyping, the identification of virulence factors and a comparison of their profiles of epigenetic modifications in the sequenced genomes. This work demonstrated that genotyping, the identification of mutations associated with antibiotic resistance and profiling of genome methylation patterns may be helpful for monitoring hospital-associated microbiota and to determine pathogens, which could potentially cause outbreaks of hospital infections.

## 2. Materials and Methods

### 2.1. Isolation of Bacterial Pathogens and Antibiotic Resistance Detection

The isolates were obtained during 2021 in the Department of Vascular Surgery of Syzganov’s National Scientific Center of Surgery in Almaty, Kazakhstan. This study was approved by the Committee of Institutional Animal Care and Use in the Scientific Center for Anti-Infectious Drugs (SCAID), Almaty (REF: #2 from 16 September 2020). The strains collected during this study were obtained using selective media, as described previously [20,21]. A routine confirmation of species’ identities was performed using the NEFERM test kit (Erba Lachema s.r.o., Brno, Czech Republic). The stock culture was kept in a freezer at −80 °C. The disk diffusion method [22] was used to determine the antimicrobial resistance of the selected isolates to the following antibiotics: amikacin (10 μg/disc); ampicillin (10 μg/disc); amoxicillin (10 μg/disc); azithromycin (30 μg/disc); carbenicillin (100 μg/disc); cefazolin (30 μg/disc); ceftriaxone (30 μg/disc); erythromycin (10 μg/disc); imipenem (10 μg/disc); gentamycin (30 μg/disc); meropenem (10 μg/disc); oxacillin (1 μg/disc); tobramycin (30 μg/disc). Zones of growth inhibition were measured after an overnight cultivation of the bacterial cultures at 37 °C on solid CHROMagar Orientation medium (bioMérieux, Craponne, France) with Endo agar (for enterobacteria and staphylococci) or cetrimide agar (for *Pseudomonas*).

### 2.2. DNA Extraction

DNA samples were extracted from bacterial cells using PureLink Genomic DNA Kits (Publication Number: MAN0000601, Revision 2.0), following the manufacturer’s recommendations. The integrity, size, quality and quantity of the isolated genomic DNA from bacterial strains were determined using the NanoDrop 2000c spectrophotometer (Thermo Scientific, Walthamm, MA, USA) at optical wavelengths of 260 nm and 280 nm and with electrophoresis on 1% agarose gels. The DNA library was prepared and quality controlled by Macrogen (Seoul, Republic of Korea). The library size and template size distribution were checked by Agilent Technologies (Santa Clara, CA, USA) 2100 Bioanalyzer using a DNA 1000 chip. The library sample concentration was checked using a Qubit standard quantification solution.

### 2.3. Genome Sequencing and Bioinformatic Analysis

The DNA libraries were sequenced in Macrogen using the PacBio Sequel-I (Pacific Biosciences, Menlo Park, CA, USA) sequencing platform, exploiting the circular consensus sequencing (CCS) system producing HiFi (high-fidelity) DNA reads ensuring a 99.5% base call accuracy suitable for genome assembly and calling polymorphic sites [23]. The average coverage of reads per one nucleotide in different genomes was 150–200 [21]. A further processing of the DNA reads was performed using software tools, as described below, with default parameter settings if not indicated otherwise. The DNA reads were quality controlled and checked for remaining adapters using LongQC v1.2.0c [24] and assembled using Canu v2.0 [25]. Plasmid contigs were identified using Platon v1.6 [26]. The original DNA reads were mapped to the scaffolds using pbmm2 (SMRT Link v10.1.0.119588) for error correction. Consensus sequences were generated from the alignments using the gcpp Arrow algorithm (SMRT Link v10.1.0.119588) and also using a pipeline of samtools-1.10, bcftools-1.7 and vcfutils.pl utilities. The consensus sequences were annotated using the PGAP NCBI annotation robot in the process of depositing the genomic contigs at the GenBank database (Table 1). Clusters of homologous genes were identified using the program GET_HOMOLOGOUS (https://github.com/eead-csic-compbio/get_homologues/releases, accessed on 6 January 2023) [27]. Genealogical links between plasmids were estimated as distances using the numbers of shared homologous genes (Equation (1)).
(1)$$D_{ij}=1-\frac{2H}{\min\left(N_{i}+N_{j}\right)}$$
where *D_ij_* is the genealogical distance between plasmids *i* and *j*; *H* is the number of homologous genes shared by two plasmids; *N_i_* and *N_j_* are the numbers of genes found in plasmids *i* and *j*, respectively. The obtained distance matrix was visualized with the neighbor-joining (NJ) algorithm implemented in *neighbor.exe*, Phylip 4.0.

Acquired antimicrobial resistance (AMR) genes were annotated with Abricatev.1.0.1 (https://github.com/tseemann/abricate, accessed on 6 January 2023) using the NCBI AMR FinderPlus [28], GRI-CARD [17,29], ARG-ANNOT [30] and MEGARES 2.00 databases [31]. Only the genes with coverages greater than 85% were included. The search for virulence factors was carried out using the Galaxy platform (https://usegalaxy.org/, accessed on 6 January 2023) via the vfdb [18] and ecoli_vf [19] databases.

The online interactive tools of PubMLST (https://pubmlst.org/, accessed on 6 January 2023) and the Pasteur Institute (http://bigsdb.pasteur.fr, accessed on 6 January 2023) were used for multilocus sequence types (MLSTs). For the genome-based detection of serotypes of enterobacteria, SerotypeFinder 2.0 was used [32]. The SPA types were assigned using the Ridom Spa Server database (http://www.spaserver.ridom.de, accessed on 6 January 2023) [33] and the software spaTyper 1.0 of the Center for Genomic Epidemiology (https://cge.food.dtu.dk/, accessed on 6 January 2023). The SPA typing was based on a comparison of 21–30 bp long repetitive regions in *spa* genes of *Staphylococcus aureus*.

Mobile horizontally acquired genomic islands (GIs) were identified using the SeqWord Genomic Island Sniffer [34]. Sequence similarities between the GIs were estimated through a comparison of the frequencies of tetranucleotides in their compositions [35]. Possible sources of GIs were predicted by searching through the Pre_GI database [36]. Oligonucleotide patterns of GIs become more similar to the host genome pattern in a process called genome amelioration, which allows for distinguishing recent inserts in bacterial genomes from old acquisitions. This stratigraphic analysis of the layers of horizontal gene acquisition was validated in a previous work on pathogenic *E. coli* [37].

The identification of methylated nucleotides and motifs of DNA methylation was performed using programs ipdSummary and motifMaker of the package SMRT Link v10.1.0.119588, as described previously [38].

### 2.4. Data Availability

The genomes of all the microorganisms are available from the GenBank database under accession numbers shown in Table 1.

## 3. Results

### 3.1. Bacterial Isolates Used in This Study

Clinical isolates from Syzganov’s National Scientific Center of Surgery in Almaty (Kazakhstan) were tested for susceptibility to the following antibiotics: amoxicillin, ampicillin, cefazolin, imipenem, oxacillin, amikacin, azithromycin, carbenicillin, ceftriaxone, erythromycin, gentamicin, meropenem and tobramycin. Isolates of different bacterial species showing multidrug-resistant phenotypes were selected for whole-genome sequencing. Selected strains, their patterns of antibiotic resistance, identified replicons and GenBank accession numbers are shown in Table 1. Detailed information regarding the diameters of the inhibition zones around the disks with the antibiotics is available on the respective BioSample pages of the isolates at NCBI (Table 1).

### 3.2. Genotyping of Isolates

Two isolates, *E. coli* 3/145 and *E. coli* 19/278, were associated with two different MLST sequence types (STs), ST3 and ST53, respectively. Serotypes of the isolates were determined with the SerotypeFinder program based on the analysis of the sequences of *fliC* genes [32]. The serotypes were H18:O17 for *E. coli* 3/145 and H5:O75 for *E. coli* 19/278. ST3 isolates are common causative agents of colitis, diarrhea and urinary infections [13]. According to the MLST database records, only a few cases of acute colitis reported in Iraq were associated with ST53.

*Klebsiella pneumoniae* isolates 13/97 and 20/245 were found to belong to ST23 (serotype K1) and ST380 (serotype K2), respectively. These are two relative STs of *K. pneumoniae*, which differed by sequences of only two marker genes, *phoE* and *tonB*. Both STs are common agents of nosocomial infections around the world, causing sepsis, liver abscesses, meningitis and pneumonia. The serotype K1 is most common in Asia, whereas the serotype K2 is more common in Europe and North America [39,40]. Several studies have shown that K1 and K2 are more virulent than other serotypes of *K. pneumoniae* [41].

Two isolates of *Pseudomonas aeruginosa*, 16/222 and 7/157, both belong to ST308 (serotype O11), and the strain *P. aeruginosa* 9/195 belongs to ST244 (serotype O5). According to the database records, ST308 strains are commonly isolated in Europe, Africa and South America from samples of bronchial mucus, blood and urine, and also from environmental water and soil samples. ST244 isolates are also widely distributed throughout the world. They are frequently isolated from patients with cystic fibrosis, but also from blood, urine and infected soft tissue samples.

Two isolates of *Staphylococcus aureus,* 597/2 and 598, belong, respectively, to ST508 and ST30. These two STs are quite distant. They differed by sequences of five out of seven *S. aureus* MLST marker genes. The ST508 strains are widespread. They are frequently isolated from the milk of cows with mastitis, but have also been associated with nasal bacteriosis in humans, skin infections, respiratory diseases and, in several cases, with sepsis. The ST30 variant is also widespread and mostly associated with sepsis, but these strains have also been isolated from skin, wounds, nasal secretions and from the environment. SPA typing identified *spa*-types t230 and t012 for the strains *S. aureus* 597/2 and 598, respectively. According to the records in the Ridom SpaServer database [33], the *S. aureus spa*-type t012 is common among clinical isolates around the world, which are frequently associated with the MRSA phenotype [42]. The *S. aureus* t230 *spa*-type is more frequent among clinical isolates from Europe.

The MLST analysis of the *Staphylococcus epidermidis* 597 showed that it may belong to a new yet unknown ST, as the sequence of its *aroE* marker gene did not show any hits against the database sequence records. A comparison of the other six marker genes revealed a single copy variant neighborhood of this strain, which included ST759, ST59, ST81 and ST519.

*Streptococcus pneumoniae* PHRX1-2021 was identified as ST377. According to the database records, one known human isolate of the ST377 (Utah35B-ST377, serotype 35B) was isolated in 1994 in the USA. Resistance to chloramphenicol, penicillin and cefotaxime was reported for this strain according to the database record.

### 3.3. Plasmids and Genomic Islands of the Sequenced Isolates

Several selected multidrug-resistant isolates contained plasmids (Table 1), which were identified in the assemble genome contigs using PlasmidFinder 2.0 [43]. Plasmids are vehicles of antibiotic resistance and virulence determinants. An analysis of the TRA operons involved in plasmid conjugation allowed for the identification of incompatibility groups of several plasmids from *Enterobacteria*. Possible genealogical relationships between these plasmids were analyzed through the presence of homologous genes in their contents. Homologous protein coding sequences shared by different plasmids were identified with the program GET_HOMOLOGOUS. Then, the distances between plasmids were calculated in a pair-wise fashion with Equation (1). A genealogical tree was inferred using the NJ algorithm based on the resulting distance matrix (Figure 1).

Enterobacterial isolates have several plasmids, which belong to different incompatibility groups. As one bacterial cell cannot contain two plasmids of the same incompatibility group, it may be assumed that the two large plasmids #1 and #2 in *K. pneumoniae* 20/245, both identified as being in the IncF group, in fact are unassembled contigs of one larger plasmid. IncF plasmids were found in *K. pneumoniae* 13/97, *E. coli* 3/145 and *E. coli* 19/278. Approximately 50% of the sequences of the large plasmid one from the *E. coli* 3/145 was almost identical to the virulence plasmid pAR-0422-2 (CP044193.1) from *Shigella sonnei*. The other half of this plasmid showed a similarity to the plasmid AH25 (CP055256.1) from *E. coli*. It cannot be excluded that two different plasmids were artificially combined during the process of genome assembly. On the other hand, it would not be possible for the two plasmids of the same incompatibility group to exist together in one bacterial cell. A natural integration of two virulence plasmids into one large plasmid seemed to be a plausible scenario.

Both *E. coli* isolates had two other smaller plasmids belonging to IncI, and the smallest IncQ plasmid #3 was found in *E. coli* 3/145. Surprisingly, in Figure 1, the latter plasmid was grouped with the plasmid one from *S. epidermidis* 597 due to sharing several homologous genes. Another two plasmids of this strain shared homologous genes with the plasmid from *S. aureus* 597/2.

Genomic islands (GIs) in chromosomes are inserts of prophages or integrated parts of conjugative plasmids. A recent study showed that the transduction of functional genes through phages may be very efficient due to a generalized transduction mechanism when phage capsids incorporate hybrid DNA fragments comprising phage genes and fragments of bacterial core genomes [44].

GIs were detected using the program SeqWord Sniffer [34]. A comparison of tetranucleotide patterns of different GIs allows grouping them by a possible common origin, the identification of similar GIs from other microorganisms and the estimation of the relative time of their acquisition by the host microorganisms [36,37]. The results of the GI clustering and estimation of the relative insertion time are given in Figure 2.

An analysis of the genealogical relationships between the GIs demonstrated that *E. coli* and *K. pneumoniae* often exchanged their genetic materials. Noteworthy was that the *K. pneumoniae* isolates contained more recently acquired GIs than the *E. coli* isolates. It may be assumed that the *K. pneumoniae* isolates have a higher propensity to acquire GIs from distant sources, which could then be transmitted to *E. coli*. The Pre_GI database allows for the tracing down of possible directions of the GIs transmissions between microorganisms [35]. It was predicted that GI#7 in the genome of the *K. pneumoniae* 13/97 likely originated from *Enterobacter*. The identified GIs showed both k-mer and sequence similarities with the records of the GIs of *Enterobacter* in the Pre_GI database. A possible source of GI#9 of *K. pneumoniae* 20/245 containing a beta-lactamase gene could be *Serratia* sp. Patterns of oligonucleotides (k-mers) of the GIs identified in one genome are usually more similar to one another than the patterns of GIs from different strains. However, three GIs, #1, #14 and #19, of the *E. coli* 19/278, were more similar to the patterns of the GIs identified in *E. coli* 3/145. This may indicate gene exchange events between the ST3 and ST53 lineages of *E. coli*, in which ST3 (strain *E. coli* 3/145) served as a donor of the genetic material. GIs #14 and #19 contain phage genes and several virulence factors, such as *vgrG*-like genes, encoding effector transporters, one putative adhesin/hemolysin precursor gene and an *ivy* inhibitor of a vertebrate c-type lysozyme.

The GIs of the *P. aeruginosa* isolates were rather homogeneous and strain specific in terms of their oligonucleotide patterns. An alarming exception was the discovered link between GI#11 of *K. pneumoniae* 13/97, GIs #3 and #12 of *P. aeruginosa* 9/195 and GI#6 of *P. aeruginosa* 16/222 that may indicate a genetic exchange between taxonomically distant pathogens. A search through the Pre_GI database suggested that it was most likely not a direct exchange between *Klebsiella* and *Pseudomonas*, but a parallel acquisition of these GIs by the bacteria from one common source, which, according to the k-mer pattern similarity, was an unknown representative of the genus *Enterobacter*. These GIs mostly comprised phage-related and hypothetical genes. However, GI#12 of *P. aeruginosa* 9/195 contains an uncharacterized beta-lactamase and a ferrous-iron efflux pump *fieF*, protecting bacteria from iron intoxication in aerobic conditions. The latter gene was also present in GI#6 of *P. aeruginosa* 16/222.

The GIs of the isolated Gram-positive bacteria were clustered separately from the GIs of the other isolates. Compositional similarities were determined between GI#7 of *S. epidermidis* 597, GI#1 of *S. pneumoniae* PHRX1-2021 and GIs #2 of *S. aureus* 598 and 597/2. The latter two GIs of *S. aureus* contained a *sdrCDE* operon of adhesins. A gene for a putative adhesin was also found in GI#7 of *S. epidermidis* 597.

Of particular interest was GI#2 from *S. epidermidis* 597, which was found to contain a methicillin resistance genetic cassette. This GI is widespread among other known *S. epidermidis* to cause hospital infections, such as the collection strains ATCC 12228 and RP62A.

Five GIs of *S. aureus* 598 and 597/2 were homologous prophage inserts. The stratigraphic analysis (Figure 2B) showed that all these prophages were rather old ancestral insertions.

### 3.4. Distribution of Antibiotic Resistance Determinants

#### 3.4.1. Antibiotic Resistance Mutations

Generally recognized mutations rendering drug antibiotic resistance (Table 2) were identified in the sequenced genomes by searching through the public databases.

Mutation R234F in the EF-Tu prolongation factor is known to render resistance to inhibitors of protein biosynthesis, such as elfamycins [45]. The inactivation of *marR* transcriptional regulators of efflux pumps leads to the development of multidrug resistance in enterobacteria [46]. Mutations in EF-Tu and *marR* were found in all the *E. coli* and *K. pneumoniae* isolates. Additionally, the strain *E. coli* 3/145 had two mutations, E448K in the *glpT* and S352T in the *cyaA*, which conferred the resistance to fosfomycin [47]. These mutations were not found in *E. coli* 19/278 and in the *K. pneumoniae* isolates. However, the *K. pneumoniae* isolates gained another known fosfomycin resistance mutation E350Q in *uhpT* [48]. Their resistance to superoxide-generating agents, macrophage-generated nitric oxide and several antibiotics such as quinolones may be associated with mutations in the *soxR* regulator that represses the oxidative stress response controlled by *soxS* [49]. A characteristic substitution, G74R, was discovered in both the *E. coli* isolates and also in *P. aeruginosa* 16/222. Destructive mutations in efflux pump inhibitors *nalC* and *mexS* [50] and a target-modifying mutation T83I in *gyrA* were found in *P. aeruginosa* 16/222 and 7/157. The strain *P. aeruginosa* 9/195 was found to have one strain-specific mutation, L71R, in the two-component response regulator *qseB*/*pmrA*, conferring the resistance to peptide antibiotics and aminoglycosides [51].

Mutations in the gene transcription repressor unchaining activities of multiple stress response systems and efflux pumps are very common in drug-resistant microorganisms [52]. One such commonly mutated repressor is the transcriptional regulator AcrR. Multiple mutations (P161R, G164A, F172S, R173G, L195V, F197I and K201M) were found in this gene in *K. pneumoniae* 20/245. These mutations are associated with resistance to fluoroquinolone [53]. In *K. pneumoniae* 13/97, this gene was truncated with a stop codon in the middle of the coding sequence. The homologous genes in the *E. coli* and *P. aeruginosa* isolates were also rather polymorphic compared to the sequences of these proteins stored at NCBI. However, it remained unclear which of these mutations could be associated with antibiotic resistance. Other polymorphic genes in the *K. pneumoniae* genomes were porins OmpK36 and OmpK37, where multiple amino acid substitutions were found to be possibly associated with resistance to cephalosporins and carbapenems [54,55].

Mutations in the *glpT* and *murA* genes of *S. aureus* are associated with resistance to fosfomycin [46,47,56]. Both of these genes bore several common and strain-specific mutations. Another strain-specific mutation, I45M, in *grlA* of *S. aureus* 597/2, conferred this strain with resistance to ciprofloxacin [57]. No drug resistance mutations were found in *S. pneumoniae* PHRX1-2021 and *S. epidermidis* 597.

**Table 2 microorganisms-11-00323-t002:** Mutations associated with antibiotic resistance identified in CARD-RGI database.

Protein	*E. coli*		*K. pneumoniae*		*P. aeruginosa*			*Staphylococcus*		Associated Resistance
	3/145	19/278	13/97	20/245	7/157	16/222	9/195	597/2	598	
MarR	G103S, Y137H	S3N, G103S, Y137H	G103D, Y137Q	G103D, Y137Q						Broad spectrum [46]
NalC					S209R, G71E	S209R, G71E				Broad spectrum [50,58]
MexS					V73A	V73A				Broad spectrum [50,58]
GrlA								I45M		Ciprofloxacin [57]
GlpT	E448K							A100V	A100V, V213I	Fosfomycin [47,56]
CyaA	S352T									
UhpT		E350Q	E350Q	E350Q						
MurA								E291D, T396N	D278E, E291D	Fosfomycin [59]
ParC		S80I								Fluoroquinolones [60]
ParE		L416F								
GyrA		S83L, D87N			T83I	T83I				
PBP3	D350N, S357N	D350N, S357N	D350N, S357N	D350N, S357N						Beta-lactams [61]
QseB							L71R			Aminoglycosides [51]
EF-Tu	R234F	R234F	R234F	R234F						Elfamycins and other peptide synthesis inhibitors [45]
SoxR	G74R	G74R				G74R				Broad spectrum [49]

We wish to stress the readers that the polymorphic sites in the sequences were identified by aligning long PacBio reads against the genome assembly with the aim of identifying mutational commonalities between the different isolates. Individual mutations were not checked with resequencing, and some level of false-positive predictions could not be excluded.

#### 3.4.2. Antibiotic Resistance Genes

Antibiotic resistance genes were predicted using the CARD-RGI web service (https://card.mcmaster.ca/analyze/rgi, accessed on 6 January 2023). The distribution of the resistance genes between the core parts of the chromosomes, predicted GIs and the plasmid sequences is shown in Table 3.

Antibiotic resistance genes of the enterobacterial isolates were located on the plasmids and chromosomes. The gene *mphA,* conferring the resistance to azithromycin, was found on the plasmids of both strains of *E. coli*. This gene is the most common macrolide resistance gene found in commensal and clinical *E. coli* [62]. Several plasmid-born genes of *E. coli*, rendering the resistance to penams and penems, were discovered, including TEM and CTX-lactamases, several penicillin-binding target replacement proteins and two beta-lactam-binding efflux pumps), tetracycline (*tetB* and *tetR*), cephalosporins (*TEM-1*) and sulfonamide (*sul1* in 3/145 and *sul2* in 19/278).

The *K. pneumonia* isolates possessed a smaller number of drug resistance genes compared to the *E. coli* isolates. It is known that the efflux pumps and enzymatic degradation of antibiotics play a key role in the multidrug-resistant *Klebsiella* [63]. Respective genes were found in the core part of the chromosomes. The discovered resistance of these strains to beta-lactam antibiotics could be associated with the presence of *SHV-11* beta-lactamase in the strain *K. pneumoniae* 13/97 and *SHV-207* in *K. pneumoniae* 20/245, in addition to the *Escherichia*-specific *ampH* beta-lactamases found in this genome. Both of these strains possessed the genes *ompA* and *ompK37*, encoding porins. According to Domenech-Sanchez et al. [64], the expression of *ompK37* in a resistant *K. pneumonia* strain decreased its susceptibility to cefotaxime and cefoxitin. Both *K. pneumonia* isolates were shown to have the genes *fosA6* and *oqxA*, inferring the resistance to fosfomycin and quinolones, respectively [65].

*P. aeruginosa* is considered one of the most frequent causative agents of hospital-acquired pneumonia, and is responsible for one-third of incidences of urological infections as well as 20–25% purulent surgical infections. *P. aeruginosa* infections are stubborn because of the natural resistance of this pathogen to many antibiotics due to their powerful MEX/OPR efflux pump systems and because of the creation of thick biofilm layers, reducing the uptake of antibiotics [66]. Many of these mechanisms are inborn and present in all sequenced genomes of this species. No plasmids were found in the studied isolates. However, multiple horizontally acquired GIs were identified in the chromosomes (Figure 1). Despite the multiple GI inserts, the antibiotic resistance genes were mostly found in the core chromosomal regions, for example, *blaZ* beta-lactamases and aminoglycoside 3′-phosphotransferases (*aph(3*′*)-IIb*). A few additional resistance genes were identified in the GIs, one of which was the *crpP* fluoroquinolone/ciprofloxacin-modifying enzyme [67], present in all three sequenced genomes. However, in the strain *P. aeruginosa* 16/222, this gene was truncated. Other horizontally acquired genes were phosphoethanolamine transferase *arnA,* conferring the resistance to peptide antibiotics, and acetyltransferase *catB7,* decomposing chloramphenicol and other phenicols [68], both found in the GIs of *P. aeruginosa* 16/222 and in the core parts of the chromosomes of the other two *P. aeruginosa*. The gene *nalD* encoding a transcriptional repressor of the MexAB-OprM multidrug efflux system was absent in *P. aeruginosa* 16/222, whereas the strains *P. aeruginosa* 9/195 and 7/157 possessed this gene. Mutations or the complete deletion of *nalD* were reported to confer multidrug resistance [69]. Other genes likely acquired through horizontal gene transfer were multiple copies of efflux pumps *adeF*, *fosA*, *parR*, *parS* and *pmpM,* found in all the *P. aeruginosa* isolates.

The genomes of the selected Gram-positive pathogens had fewer antibiotic resistance genes. The genome of the strain *S. epidermidis* 597 comprised 18 such genes. Ten of them were located on the chromosome, five on plasmid two and three on plasmid three. The most notorious was the *mecA* gene located in the pathogenicity of GI#1, rendering the methicillin resistance in MRSA *Staphylococci*. Other important antibiotic resistance genes of this strain were beta-lactamase *blaZ* [70]; *msrA,* which protects ribosomes from toxic compounds; macrolide 2′-phosphotransferase *mphC*; and multidrug efflux pump *norA* and *fosAB,* conferring resistance to fosfomycin. The gene *fosAB* was also found in *S. aureus* 598, but not in *S. aureus* 597/2. Other antibiotic resistance genes found in these genomes were the tetracycline resistance gene *tet38* and the norfloxacin/ciprofloxacin resistance gene *mepA*. The *S. aureus* isolates had no inserts of the SCCmec cassette with the *mecA* methicillin resistance genes, even that the strain *S. aureus* 597/2 was predicted to be a t012 *spa*-type, which is associated with MRSA isolates. The absence of SCCmec was confirmed by mapping the PacBio reads generated for these microorganisms against the SCCmec sequence from the reference genome CP033505.1 [38].

The strain *S. pneumoniae* PHRX1-2021 possessed the smallest number of antibiotic resistance genes. Only three efflux pumps (*pmrA*, *patB* and *patA*) and one ribosomal methyltransferase (*rlmA(II)*) were found. The latter gene conferred the resistance to mycinamicin, tylosin and lincosamides. These antibiotic resistance genes were located in the core parts of the chromosome.

### 3.5. Distribution of Virulence Genetic Determinants

Virulence genes were identified in the sequenced genomes by searching through the VFDM database (http://www.mgc.ac.cn/VFs/, accessed on 6 January 2023) [18]. Identified virulence genes and their localization in the chromosomes, plasmids and GIs are shown in Table 4. The full list of virulence genes identified in the selected genomes is given in Appendix A.

#### 3.5.1. Virulence Factors of *E. coli*

The majority of the virulence factors of extraintestinal pathogenic *E. coli* (ExPEC) control biofilm formation. They include various adhesins, toxins, iron absorption factors, lipopolysaccharides and capsular polysaccharides biosynthetic genes [71]. ExPEC possesses specific adhesive factors that allows it to colonize host tissues where *E. coli* does not normally occur, such as the small intestine and urethra. An important group of virulence factors of *E. coli* associated with pyelonephritis and recurrent cystitis is the family of *afa/dra* adhesins, which contain both fimbrial and afimbrial adhesins. The strains *E. coli* 3/145 and 19/278 had *afaB*, *afaC* and *afaF* adhesins, characteristic of uropathogenic *E. coli* (UPEC). The strain *E. coli* 3/145 had another adhesin *draB,* also associated with urinary tract infections. The adhesin *fimH* was present in both *E. coli* isolates. This adhesin aids in binding to the uroplakin 1A receptor (UP1a) of epithelial cells of the bladder to allow for the colonization of infection spots [72,73]. Both of these strains had extra copies of *fimE* adhesins located in the plasmids. Other groups of virulence factors encoding S fimbriae adhesins (*sfa*/*foc*) and P-like pili (*papC* and *Iha*), which support the bacterial colonization of the urinary tract [74], were also found in these two isolates. It can be assumed that these isolates are associated with urinary tract infections.

More than 40 genes involved in iron scavenging were identified in sequenced *E. coli*. In addition to four operons of aerobactin (*iut*A/*iuc*ABCD), yersiniabactin (*ybt*), enterobactin (*ent*) and salmochelin (*iro*BCDEN), genes for iron/manganese transport (*sit*ABCD) and heme uptake (*chu*) were also found. The strain *E. coli* 19/278 had an extra copy of *shuA* in its large plasmid.

The discovery of NMEC virulence factors in the strains *E. coli* 3/145 and 19/278 is of importance, as these genes encode the K1 capsular antigen that protects bacterial cells from phagocytosis. Other important virulence factors are the gene *ibeBC,* supporting brain endothelial cell invasion; *iss,* which protects pathogenic cells from phagocytosis; and *traT,* which suppresses the immune response of the host. Interestingly, all of these factors are not common in regular UPEC isolates [73]. This may indicate a trend towards generalizing infections caused by these isolates.

#### 3.5.2. Virulence Factors of *K. pneumoniae*

In the sequenced *K. pneumoniae* isolates, many virulence factors were plasmid-born. Large plasmids of these strains were homologous to the virulence plasmid pLVPK from *Klebsiella* (AY378100.1). These plasmids contain multiple virulence genes, including *rmpA*, *ibeB*, *iha*, *iroB*, *iroN*, *irp*, *iucA* and *iutA* [75]. Hypervirulent *K. pneumoniae* (hvKP) has spread worldwide, since this variant was first detected in the Asia-Pacific region. This is an invasive variant with an enhanced capsule production and hypervirulence that differs from nonvirulent *K. pneumoniae* (cKP). The main factors contributing to hvKP include multiple siderophores, capsular polysaccharide production, lipopolysaccharides (LPS) and fimbria [76,77].

Four typical siderophores of *Klebsiella*, enterobactin, yersiniabactin, salmochelin and aerobactin, which are common for hvKP and less common for cKP [78], were found in the sequenced isolates. In both strains, extra copies of *iutA*-encoding aerobactin, *irp-* and *fyuA*-encoding yersiniabactin and its receptor, *iroB*/*iroD* (involved in salmochelin synthesis) and hvKP-specific iron transporter, *kfu* [79], were found in the chromosomes and/or plasmids of the isolates. Both strains possessed operons of 18 genes encoding the synthesis of the toxic metabolite colibactin (*clbA-R*, a colorectal cancer-provoking toxin) that damages eukaryotic DNA, facilitates mucous membrane colonization and contributes to the spread of pathogens through the bloodstream [80].

Another virulence factor common for hvKP found in both sequenced *Klebsiella* isolates was the type six secretion system (T6SS). T6SS works like a nanosyringe that injects effector proteins, destroying the cells of the host [81]. The obtained data suggests that the clinical isolates *K. pneumoniae* 20/245 and 13/97 belong to hvKP, and may cause infections in hospitals and communities.

#### 3.5.3. Virulence Factors of *P. aeruginosa*

It is generally recognized that *P. aeruginosa* is one of the most common causes of ventilator-associated pneumonia (VAP), due to possessing an arsenal of virulence factors. The clinical isolates of *P. aeruginosa* selected for this study were equipped with 375 common virulence genes and several strain-specific genes. The biggest number of virulence genes (377) was observed in the strain *P. aeruginosa* 9/195.

Many virulence factors are associated with the development of VAP. In addition to biofilm formation, the main virulence factors of *P. aeruginosa* include elastase, phospholipase C, protease A, exotoxins and cytotoxins, flagella and pili proteins, pigment production and quorum-sensing (QS) proteins [82]. The two sequenced *P. aeruginosa* strains 7/157 and 16/222 had type III toxin-encoding genes, *exoU*, *exoT* and *exoY*. According to Feltman et al. [83], *exoT* is constituent in all clinical and environmental *P. aeruginosa* isolates, whereas *exoS* is found in approximately 70% of clinical isolates. The toxin ExoU is the most virulent factor among *P. aeruginosa*’s type III effectors, which was found to be present in approximately 30% of clinical isolates. The strain *P. aeruginosa* 9/195 had an additional effector gene, *exoS*, but did not have *exoU*. Biofilm formation due to *P. aeruginosa* occurs along with the production of various extracellular components, such as type IV pili, fimbria, exopolysaccharides, adhesins, LecA/LecB lectins and Fap amyloids [84]. A large number of the virulence genes identified in the sequenced genome (over 70) were related to the synthesis of flagella and type IV pili proteins. Additionally, the genes of the two QS systems, *las* and *rhl*, were found in these strains. In addition, more than 20 genes were associated with alginate biosynthesis, a polysaccharide contributing to the persistence of bacteria in the lungs [85]. Another important virulence factor is *algU,* encoding an extracytoplasmic sigma factor that plays a key role in cystic fibrosis pathogenesis through *P. aeruginosa* [86]. All three selected *P. aeruginosa* strains had *plcH* gene-encoding hemolytic phospholipases C, which is an important virulence factor of these pathogens [87].

Siderophores, particularly pyoverdine and pyocyanin, are essential for the survival of *P. aeruginosa* in the host where free iron is limited. Pyochelin has a lower affinity with iron than pyoverdine, and is believed to be associated with the persistent inflammatory response seen during chronic infections [88]. All the genes involved in the synthesis of pyoverdine, pyocyanin and pyochelin were found in the investigated strains. An interesting discovery was the presence of the genes *ybtP* and *irp,* involved in the synthesis of yersiniabactin, that are more commonly produced by the phytopathogenic *Pseudomonas syringae* [89].

#### 3.5.4. Virulence Factors of the Gram-Positive Isolates

Relatively fewer virulence genes were identified in the strains of Gram-positive pathogens, and many of them were located in the core part of the chromosomes (Table 4).

The colonization of the host tissues by *S. aureus* starts with a surface adhesion supported by several adhesins. Typical members of the family of adhesins are the staphylococcal protein A (*spA*), fibronectin-binding proteins (*fnbAB*), the collagen-binding protein (*cna*) and gluing factor proteins (*clfAB*) [90]. Both strains, *S. aureus* 597/2 and 598, had all of these adhesins in their genomes, in contrast to *S. epidermidis* 597, which only had the genes *clfB* and *fnbB*.

Many pathogenic *S. aureus* secrete various endo- and exotoxins, among which α-hemolysin, β-hemolysin, γ-hemolysin, leukocidin, toxic shock syndrome toxin-1 (TSST-1) and several staphylococcal enterotoxins (*sea*, *seb*, *secN*, *sed*, *see*, *seg*, *seh* and *sei*) are the most common [91]. Genes for α- and γ-hemolysins were not found in the genomes of the strains *S. aureus* 597/2 and 598. However, they had *hld*-encoding δ-hemolysin and also several genes for β-hemolysins and cytolysins, including *cylG*, *cylZ*, *cylA*, *cylI* and *cylF*. Multiple copies of superantigen-like proteins (SSLs) were found in these genomes. Several additional SSL genes, *sea* and *selQ*, were detected in the plasmid of the strain *S. aureus* 597/2. The strain *S. epidermidis* 597 only had several genes involved in the synthesis of the staphylococcal superantigen-like (SSL) proteins, including *cylG*, *cylZ*, *cylA* and *cylI*.

Other *S. aureus* virulence factors inhibiting the immune system of the host are the staphylococcal complement inhibitor (SCIN), chemotaxis inhibitory protein (CHIPS) and staphylokinase (SAK) [92]. All these genes were present in the studied genomes of *S. aureus*, but these genes were not found in *S. epidermidis* 597.

Staphylococcal iron-scavenging proteins are encoded by the genes *isd* and *luc*. *S. epidermidis* 597 has extra copies of *sitB* in its plasmid. These genes are functionally related to proteins controlling the synthesis and transportation of siderophores in other bacteria.

The smallest number of virulence factors was found in *S. pneumoniae* PHRX1-2021. These genes included *cps* and *cap*, which are involved in the synthesis of polysaccharide capsules. Several other found genes (*pspA*, *pspC*/*cbpA*, *pavB*/*pfbB*, *pce*/*cbpE*, *scpA*/*scpB* and *pavB*/*pfbB*) contribute to the adhesion and colonization of the respiratory airways [93,94,95,96]. Other identified virulence factors were the *lytA* and *cbpD* autolysins, which cleave peptidoglycan and release several pneumococcal antigens, including pneumolysin, peptidoglycan and teichoic acids [97]. The release of these antigens causes serious inflammation and inhibits cytokine production, which, in turn, inhibits the activation of phagocytes. Other virulence factors comprised membrane-acting hemolytic toxin *slo*, *pvl* Panton–Valentine leukocidin and several β-hemolysins, including *cylA*, *cylG*, *cylI* and *cylZ*.

### 3.6. Genome Methylation Patterns and Restriction–Modification Systems

Motifs of methylated nucleotides were predicted for all the sequenced genomes (Table 5). The methylation of nucleotides is controlled by methyltransferases, which are often parts of restriction–modification (RM) systems, but may act as individual ‘orphan’ methylases. Methyltransferases bind to specifically recognized DNA motifs and often methylate DNA molecules at both strands. Genes encoding methyltransferases and whole RM systems are often associated with mobilomes. This is why the patterns of DNA methylation may vary between different strains of the same species.

The bipartite methylation of adenosine residues at the sixth carbon atom (m6A-methylation) in the four-nucleotide-long palindrome GATC is one of the most frequent types of epigenetic modifications of bacterial chromosomes. This methylation is controlled by a family of orphan Dam methyltransferases. The involvement of GATC methylation in the enhanced persistence of pathogens [98], the expression of antibiotic resistance genes [99] and general stress response [100] was demonstrated experimentally. This type of methylation and the respective genes for Dam methyltransferases were discovered in all the sequenced isolates of *E. coli* and *K. pneumoniae*, but also in the phylogenetically distant *S. pneumoniae* PHRX1-2021. GATC methylation is common for *Streptococcus,* and several Dam-type methyltransferases for this genus can be found in the GenBank database. Surprisingly, the BLASTP alignment of the Dam methyltransferase of the strain *S. pneumoniae* PHRX1-2021 showed to only match with the homologous genes from *E. coli*. This finding exemplified that methyltransferases may be exchanged even between Gram-positive and Gram-negative bacteria.

In several cases, more than one methylation motif could be discovered per genome. DNA motifs are specific for different methyltransferases, creating unique strain-specific patterns of DNA methylation. Due to an interference in the methylated nucleotides with gene expression regulation, the acquisition of a new RM system within a mobile genetic element may affect the transcription of many genes at once. For example, an additional DNA methylation motif, **A**GCNNNNNC*T*TC, was found in *E. coli* 3/145 (thereafter in the methylation motifs, the methylated nucleotide was shown in bold, and the nucleotide opposing the methylation site on the reverse complement strand was italicized). This methylation was associated with the activity of the type I RM system *Eco*KI. Figure 3 shows the distribution of the methylated sites **A**GCNNNNNC*T*TC and GA**A**GNNNNNGC*T* in the chromosome of this strain. In total, 598 methylated sites were found in *E. coli* 3/145, which were located within the coding sequences of 164 genes and in the promoter regions of seven genes, including ferric enterobactin-binding periplasmic protein *fepB*, curli production assembly/transport component *csgG*, lactate-responsive regulator *lldR*, an uncharacterized glutamate racemase and three hypothetical proteins. None of these sites were methylated in *E. coli* 19/278.

The genomes of the *P. aeruginosa* 7/157 and 16/222 were methylated at motifs C**A**GNNNNN*T*GGG. These genomes had type I and type III RM systems and an orphan type I methyltransferase comprising S and M subunits. An alternative methylation motif G**A**CNNNNN*T*GCC was predicted in *P. aeruginosa* 9/195. This genome comprised only a single type I RM system. This change in the methylation patterns affected many genes. In total, 687 sites were methylated in the strain *P. aeruginosa* 16/222. Methylated nucleotides were present in the coding sequences of 268 genes and in the promoter regions of 18 genes, including an iron siderophore receptor protein, Cu-oxidizing cytochrome *cbb3*, efflux pump *emrA* and general secretion pathway protein G. In the strain *P. aeruginosa* 9/195, in total, 1416 sites were methylated, which fell into the coding sequences of 555 genes and promoter regions of 31 genes involved in metabolism and one type VI secretion system component, *icmF*. These differences between methylation patterns could affect the gene expression in these pathogens.

Three methylation motifs were identified in *S. pneumoniae* PHRX1-2021. In addition to the *E. coli*-type Dam methyltransferase, this genome comprised two type II RM systems *Mja*III and *Xba*I, and seven type I RM systems; however, many of these RM systems appeared truncated. It was not clear which of these type I systems remained active, as only one type I motif was found among the methylated sites (Table 5). This example showed that methylation patterns may change either due to the acquisition of new methyltransferases or the inactivation of available RM systems. This concept was also exemplified in the sequenced *S. aureus* genomes. Both genomes had two type I RM systems. However, only one methylation motif was identified in the strain *S. aureus* 597/2, and two alternative motifs in *S. aureus* 598. Eventually, 360 sites were methylated at both DNA strands in the strain *S. aureus* 597/2, and 1168 methylated sites were found in the strain *S. aureus* 598. The strain 598 had a gene for McrBC 5-methylcytosine endonuclease, which was absent in the strain 597/2. This enzyme cleaves DNA at cytosine residues methylated at their fifth carbon atoms. This type of methylation is more specific for eukaryotes, and this enzyme could probably be added to the list of virulence factors of this strain. However, this hypothesis has not yet been proven experimentally.

One DNA methylation motif, GGTG**A**, was identified in *S. epidermidis* 597. It corresponded to the type IIS *Hph*I RM system. In contrast to the methylation motifs of the other genomes, this nonpalindromic sequence was methylated only at one DNA strand. This methylation may have only affected those genes, which were transcribed from the same DNA strand where the methylated adenine residues were located.

## 4. Discussion

With this study, an attempt was undertaken to identify the most important properties of phylogenetically diverse pathogenic bacteria causing hospital infections. It was noteworthy that the different Gram-positive and Gram-negative pathogens isolated from the same hospital environment developed similar patterns of antibiotic resistance (Table 1). The fastest way to develop resistance against antimicrobials persisting in the environment is to accumulate drug resistance mutations, which transform the target molecules or unchain the expression of a broad range of efflux pumps. The contamination of the environment with antimicrobials benefits the bearers of these mutations, and they become abundant in the bacterial populations. Table 2 shows that mutations of this type were identified in the selected pathogens belonging to different taxonomic groups. It could be expected because all these bacteria living in the same environment were under the pressure of the same antimicrobials. Searching for common drug resistance mutations in the sequenced genomes or metagenomes may be instrumental for monitoring the level of contamination in hospitals or environmental samples with antibiotics and disinfectants.

Further steps of the virulence development are associated with the horizontal acquisition of various antibiotic resistance and virulence determinants (Table 3 and Table 4). The propensity to cause hospital infection outbreaks due to a lineage of pathogens is predefined in their genotypes. All types of genotyping, including MLST, SPA and serotyping, are helpful for monitoring the epidemiological situation and for estimating the likelihoods of disease outbreaks. The number of strains used in this study was not sufficient to conclude which pathogen STs were the most abundant in the Syzganov’s National Scientific Center of Surgery in Almaty (Kazakhstan). However, even this small collection demonstrated that the bacterial population associated with hospital environments may be quite versatile, as the isolates of the same species belonged to different STs and were characterized with strain- and ST-specific repertoires of factors of pathogenicity.

All the studied isolates were abundant with broad spectrum efflux pumps, which increase the general unspecific levels of resistance to various antibiotics and disinfectants. These genes were mostly found in the core parts of the chromosomes, but also in the GIs and the plasmids. The selected strains of *E. coli* and *P. aeruginosa* possessed the highest number of antibiotic resistance genes per genome—more than 50. They were followed by the *K. pneumoniae* isolates. Only 4–5 antibiotic resistance genes were found in the Gram-positive isolates. It should be noted that these results may not be comprehensive, as only those genes were identified, which were present in the CARD-RGI database. For example, an uncharacterized beta-lactamase was predicted in GI#12 of *P. aeruginosa* 9/195, but this gene was not detected in the search through the CARD-RGI. Anyway, it may be concluded from this study that the genes encoding the recognized antibiotic resistance proteins, including several beta-lactamases, were mostly found in the core part of the genomes. The numbers of plasmid-born and horizontally acquired antibiotic resistance genes were relatively low. Interestingly, *Klebsiella*-specific genes for *kpn* efflux pumps were also found in the sequenced *E. coli*, whereas *Escherichia*-specific *ampH* beta-lactamases were found in the sequenced *Klebsiella*. These findings may be indicative of a cross-species transfer of drug resistance factors.

Different taxa of the sequenced microorganisms were characterized with different numbers of identified virulence factors. However, this may reflect a rather biased representation of pathogenic species in the VFDM database. For example, the database records of virulence factors of *Enterobacteria* and *Pseudomonas* include many housekeeping genes, whereas the virulence factors of other pathogens are scarcely represented in this database.

Interesting information could be gained by conducting an analysis of the distribution of the identified virulence factors between the conservative parts of the chromosomes and the mobile genetic elements (Table 4). It was found that almost 50% of the virulence factors identified in the *E. coli* isolates were located in their GIs and plasmids. It may be concluded that the virulence potential of these microorganisms relied to a large extent on the horizontally acquired genes. In the sequenced *K. pneumoniae* genomes, approximately 30% of the virulence factors were located in the plasmids and GIs. As to the *P. aeruginosa* and the Gram-positive isolates, there were less than 10% of the virulence genes associated with mobilomes, and none of them was in *S. pneumoniae* PHRX1-2021. Attention was paid to the distribution of antibiotic resistance and virulence genes between the core and mobile parts of the genomes due to a hypothesis stating that mobilome-associated genes may have a greater effect on the phenotype than the genes located in the core chromosomal regions [13].

Profiling the methylated sites (methylomics) in the sequenced genomes could be instrumental to enforcing the used genotyping systems, as different lineages of the same species often had quite different methylation patterns (Table 5). Moreover, the acquisition of a new RM system by a bacterium or the inactivation of available RM systems may affect many phenotypic properties of pathogens by affecting the gene expression control. This phenomenon is known as phase-variable regulation [101]. The identification of active RM systems by analyzing genome-wide methylation patterns may aid in the estimation of risks surrounding new emerging superbugs of future hospital infection outbreaks. A hypothesis was suggested in the literature that the presence of active RM and/or CRISPR-Cas systems in a bacterial genome might prevent the distribution of virulence-associated plasmids and mobile genetic elements [102]. The discoveries of this study, in general, supported this hypothesis. GIs bearing multiple virulence factors (Table 4) were found in the genomes of *Escherichia* and *Klebsiella*, in which only GATC methylation was identified. This methylation was controlled by the orphan Dam methyltransferases, which were not parts of functional RM systems. Notably, many of these GIs were recently acquired (Figure 2B). However, it was found that the strain *E. coli* 3/145 with an additional active *Eco*KI-like RM system had a smaller number of horizontally transferred virulence genes than the strain *E. coli* 19/278 (Table 4). It may be assumed that the latter strain could potentially be more dangerous as a cause of nosocomial infections. In a similar way, the strain *S. aureus* 597/2 may pose a higher risk of disease outbreaks than the strain *S. aureus* 598, because of its degenerated methylation pattern and, as a result, a bigger number of virulence genes. The genome of *P. aeruginosa* 9/195 was more heavily methylated than the two other genomes of *P. aeruginosa*, which resulted in finding a smaller number of horizontally transferred virulence genes in this bacterium (Table 4).

The methylation of chromosomal nucleotides interferes with gene expression. Therefore, it may be assumed that a significant level of methylation of bacterial genomes may restrict their ability to adapt to new conditions and environments. This hypothesis needs a proper statistical validation in further studies.

## 5. Conclusions

In this study, several pathogens representing distant taxonomic groups, all characterized with similar antibiotic resistance patterns, were isolated from the same clinic environment and used for whole-genome sequencing and a comparison. Being under pressure from the same selection forces, the drug resistance development in these bacteria started with the accumulation of specific mutations, many of which are common for phylogenetically distant pathogens. Generally, it could be concluded that third-generation sequencing, i.e., SMRT PacBio, provides bacteriologists with a high-throughput tool to monitor hospital infections. The potential of the clinical isolates to cause outbreaks of hospital infections can be estimated by paying attention to the number of virulence and antibiotic resistance genes in mobile genetic elements (GIs and plasmids), the frequency of polymorphisms in house-keeping genes associated with antibiotic resistance and to the frequency and patterns of nucleotide methylation reflecting the presence of active RM systems. Genotyping and patterns of genome methylation could also be instrumental for monitoring the distribution of individual clonal lines of pathogens due to the lineage specificity of methylation patterns.

## Figures and Tables

**Figure 1 microorganisms-11-00323-f001:**
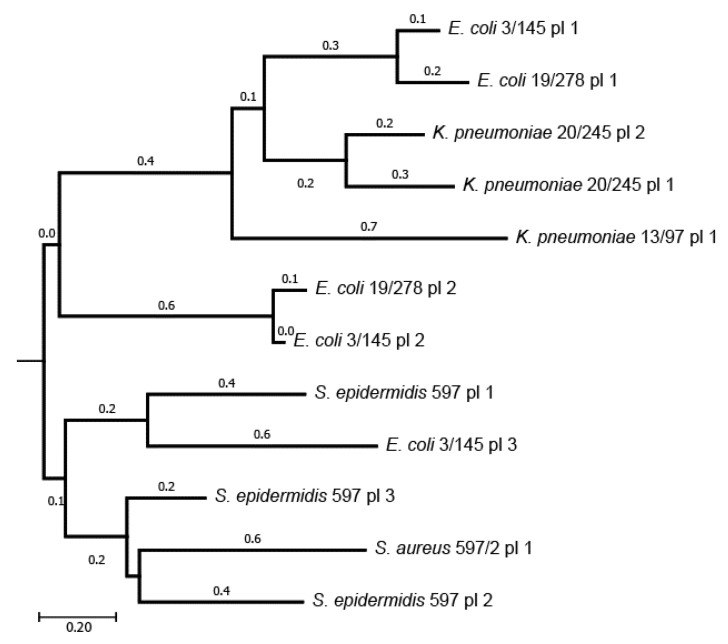
Genealogical tree of plasmids of the selected isolates.

**Figure 2 microorganisms-11-00323-f002:**
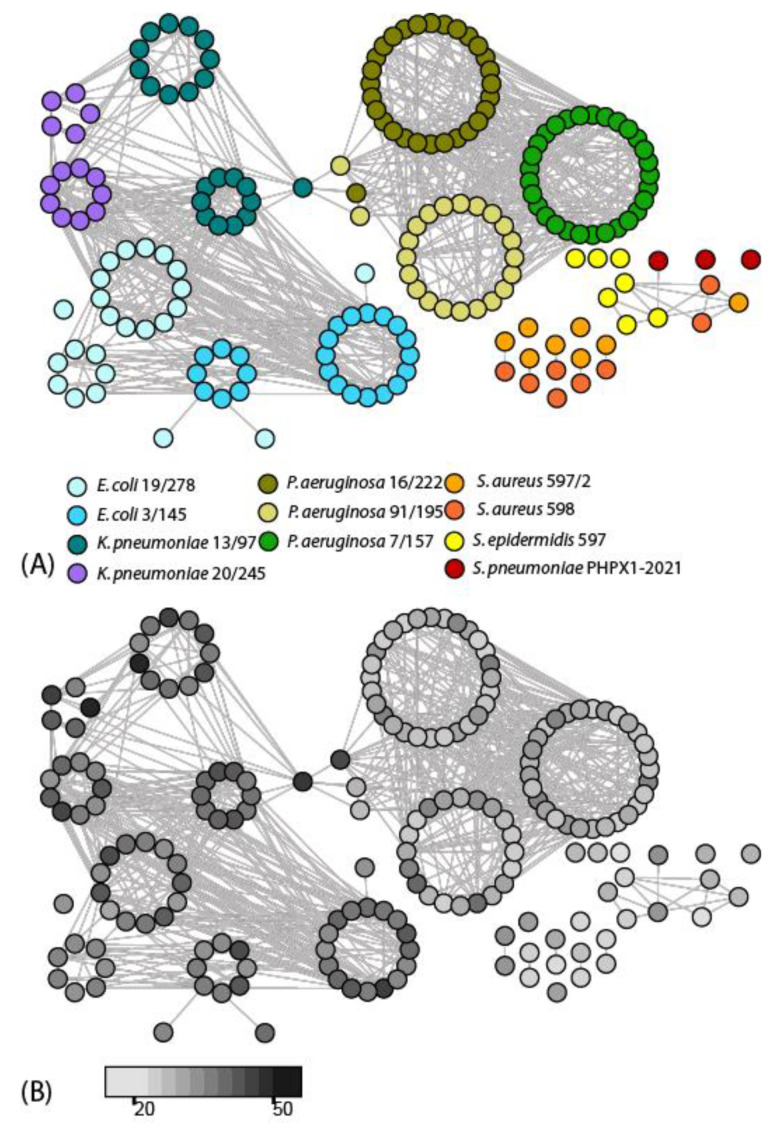
(**A**) Clustering of GIs identified in different genomes of pathogenic isolates through similarities in their patterns of tetranucleotides. Each node on the scheme corresponds to one GI. Numbers in several nodes are the consecutive clockwise numbers of GIs identified in the genomes, starting from the replication origins. Host microorganisms are depicted using different colors, as explained in the legend. (**B**) The network of GIs is identical to that in part A, but recolored in a way that the darker grey color represents more recent inserts of GIs, i.e., the GIs with more dissimilar patterns of tetranucleotides compared to the patterns calculated for the host microorganisms. The gradient bar below the scheme shows the correspondence of the grey shading to the distance between patterns calculated with the SeqWord Sniffer program.

**Figure 3 microorganisms-11-00323-f003:**
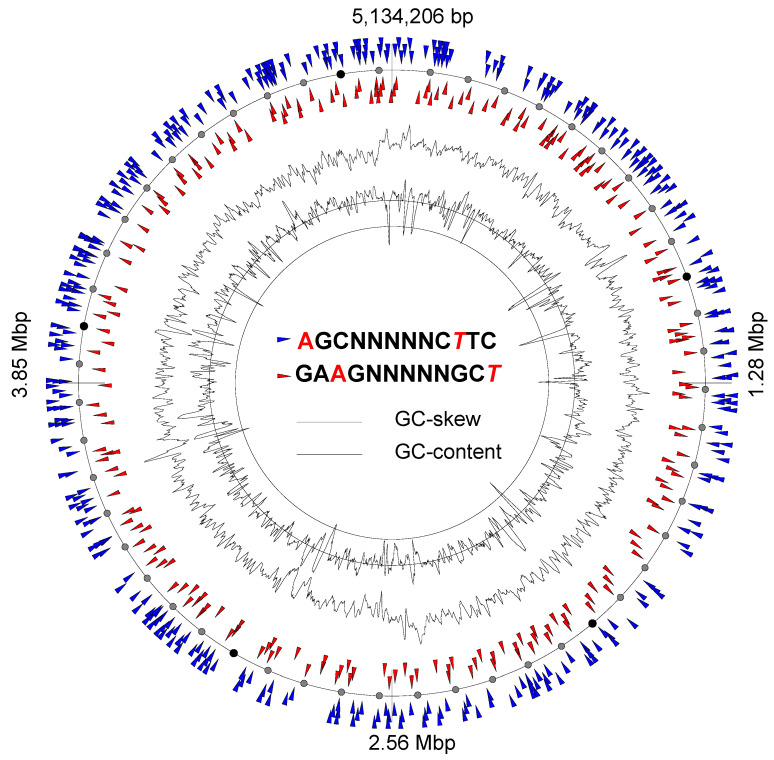
Atlas presentation of the circular chromosome of the strain *E. coli* 3/145. Locations of motifs methylated at both DNA strands are depicted with colored triangles, as explained in the legend.

**Table 1 microorganisms-11-00323-t001:** Bacterial isolates used in this study.

Strain Name	Resistance	Isolation Metadata	Replicons	Length (bp)	GenBank Acc
*Escherichia coli* SCAID WND2-2021 (3/145)	AMP, AMX, AZM *, E, OXA	Swabbed wound discharge	Chromosome	5,134,206	CP082827
			Plasmid 1	139,267	CP082828
			Plasmid 2	106,249	CP082829
*Escherichia coli* SCAID URN1-2021 (19/278)	AMP, AMX, CEF, CTR, E, OXA	Urine, patient with pyelonephritis	Plasmid 3	32,040	CP082830
			Chromosome	5,168,688	CP082824
*Klebsiella pneumoniae* SCAID PHRX1-2021 (13/97)	AMP, AMX, AZM, E, OXA	Swab from pharynx, patient with pneumonia	Plasmid 1	108,070	CP082825
			Plasmid 2	86,164	CP082826
			Chromosome	5,498,275	CP082805
			Plasmid	217,781	CP082806
*Klebsiella pneumoniae* SCAID PHRX2-2021 (20/245)	AMP, AMX, E, OXA	Swab from pharynx, patient with pneumonia	Chromosome	5,319,600	CP082796
			Plasmid 1	162,135	CP082797
			Plasmid 2	95,203	CP082798
*Pseudomonas aeruginosa* SCAID TST1-2021 (7/157)	AMP, AMX, AZM *, CEF, CTR *, E, IPM, OXA	Swab from a tracheostomy tube after an operation	Chromosome	7,173,620	CP082823
*Pseudomonas aeruginosa* SCAID PLC1-2021 (16/222)	AMP, AMX, AZM, CEF, E, OXA	Pleural cavity during an operation	Chromosome	7,124,329	CP082821
*Pseudomonas aeruginosa* SCAID WND1-2021 (9/195)	AMP, AMX, CTR, CEF, E, IPM *, OXA	Swabbed wound discharge	Chromosome	7,093,992	CP082822
*Streptococcus pneumoniae* SCAID PHRX1-2021	AMP, E, OXA	Swab from pharynx, patient with pneumonia	Chromosome	2,098,200	CP082820
*Staphylococcus epidermidis* SCAID OTT1-2021 (597)	AMP, AMX, AZM, E	Otitis, swab from ear	Chromosome	2,099,244	CP082816
			Plasmid 1	24,456	CP082817
			Plasmid 2	24,520	CP082818
			Plasmid 3	13,203	CP082819
*Staphylococcus aureus* SCAID OTT1-2021 (597/2)	AMP, AMX, OXA	Otitis, swab from ear	Chromosome	2,737,085	CP082813
			Plasmid 1	33,923	CP082814
*Staphylococcus aureus* SCAID WND1-2021 (598)	AMP, AMX	Swabbed wound discharge	Chromosome	2,889,511	CP082815

* An intermediate resistance was recorded for these antibiotics. Abbreviations: AMP—ampicillin; AMX—amoxicillin; AZM—azithromycin; CEF—cefazolin; CTR—ceftriaxone; E—erythromycin; IPM—imipenem; OXA—oxacillin.

**Table 3 microorganisms-11-00323-t003:** Numbers of antibiotic resistance genes of different categories detected in the studied genomes on the core parts of the chromosomes (Chrs), identified genomic islands (GIs) and on the plasmids (Pls).

Strain	GeneLocation	Type of Antibiotic Resistance Mechanism					
		Beta- Lactamases	Antibiotic Inactivation	Target Alteration	Target Replacement or Protection	Efflux Pump	Reduced Permeability to Antibiotic
*E. coli* 3/145	Chr	3	0	6	0	30	1
	GIs	0	0	1	0	6	0
	Pl	1	3	1	1	2	0
*E. coli* 19/278	Chr	2	0	5	0	35	1
	GIs	0	0	1	0	4	0
	Pl	2	2	1	2	1	0
*K. pneumoniae* 13/97	Chr	2	1	2	0	13	1
	GIs	0	0	0	0	3	0
	Pl	0	0	0	0	0	0
*K. pneumoniae* 20/245	Chr	1	2	2	0	15	3
	GIs	0	0	0	0	0	0
	Pl	0	0	0	0	0	0
*P. aeruginosa* 7/157	Chr	1	1	6	0	41	2
	GIs	0	4	0	0	1	0
*P. aeruginosa* 9/195	Chr	1	3	6	0	43	0
	GIs	0	2	0	0	2	2
*P. aeruginosa* 16/222	Chr	1	2	4	0	41	2
	GIs	0	2	1	0	0	0
*S. pneumoniae* PHRX1-2021	Chr	0	0	1	0	3	0
	GIs	0	0	0	0	0	0
*S. epidermidis* 597	Chr	1	1	0	0	1	0
	GIs	0	0	0	1	0	0
	Pl	0	0	0	0	0	0
*S. aureus* 597/2	Chr	0	0	0	0	5	0
	GIs	0	0	0	0	0	0
	Pl	0	0	0	0	0	0
*S. aureus* 598	Chr	1	1	0	0	4	0
	GIs	0	0	0	0	0	0

CHR—chromosome; GIs—genomic islands; Pl—plasmid.

**Table 4 microorganisms-11-00323-t004:** Number of virulence factors identified in the core parts of the chromosomes, in GIs and on the plasmids.

Strain	Core Chromosome	GIs	Plasmids
*E. coli* 3/145	131	GI#1: 3 genes; GI#4: 14 genes; GI#5: 1 gene; GI#6: 6 genes; GI#9: 3 genes; GI#11: 5 genes; GI#13: 3 genes; GI#14: 3 genes; GI#15: 3 genes; GI#16: 1 gene; GI#17: 2 genes; GI#19: 7 genes; GI#20: 8 genes; GI#21: 1 gene; GI#23: 3 genes; **TOTAL**: 63	pl_1: 7 genes; pl_2: 6 genes; pl_3: 1 gene; **TOTAL**: 14
*E. coli* 19/278	122	GI#1: 3 genes; GI#2: 2 genes; GI#4: 28 genes; GI#5: 5 genes; GI#9: 4 genes; GI#11: 11 gene; GI#12: 3 genes; GI#13: 2 genes; GI#14: 3 genes; GI#17: 7 genes; GI#18: 4 genes; GI#19: 2 genes; GI#20: 9 genes; **TOTAL**: 83	pl_1: 7 genes; pl_2: 5 genes; **TOTAL**: 12
*K. pneumoniae* 13/97	111	GI#5: 24 genes; GI#6: 7 genes; GI#7: 7 genes; GI#8: 8 genes; GI#9: 2 genes; GI#14: 11 genes; GI#21: 2 genes; **TOTAL**: 61	pl: 23 genes; **TOTAL**: 23
*K. pneumoniae* 20/245	122	GI#1: 2 genes; GI#4: 17 genes; GI#5: 7 genes; GI#6: 7 genes; GI#7: 8 genes; GI#13: 2 genes; **TOTAL**: 43	pl_1: 21 genes; pl_2: 2 genes; **TOTAL**: 23
*P. aeruginosa* 7/157	352	GI#1: 2 genes; GI#2: 2 genes; GI#4: 1 gene; GI#6: 6 genes; GI#10: 1 gene; GI#17: 2 genes; GI#19: 1 gene; GI#20: 3 genes; GI#21: 2 genes; GI#26: 2 genes; GI#27: 1 gene; GI#28: 1 gene; **TOTAL**: 24	NA
*P. aeruginosa* 9/195	358	GI#2: 1 gene; GI#5: 1 gene; GI#6: 9 genes; GI#16: 1 gene; GI#17: 2 genes; GI#18: 5 genes; **TOTAL**: 19	NA
*P. aeruginosa* 16/222	351	GI#1: 2 genes; GI#2: 2 genes; GI#5: 1 gene; GI#6: 1 gene; GI#7: 7 genes; GI#17: 2 genes; GI#19: 1 gene; GI#20: 2 genes; GI#24: 5 genes; GI#28: 1 gene; **TOTAL**: 24	NA
*S. pneumoniae* PHRX1-2021	50	0	NA
*S. epidermidis* 597	35	GI#1: 1 gene; GI#2: 1 gene; GI#6: 2 genes; **TOTAL**: 4	pl_1: 2 genes; **TOTAL**: 2
*S. aureus* 597/2	108	GI#1: 3 genes; GI#6: 1 gene; GI#7: 3 genes; **TOTAL**: 7	pl: 2 genes; **TOTAL**: 2
*S. aureus* 598	102	GI#1: 3 genes; GI#2: 2 genes; **TOTAL**: 5	NA

**Table 5 microorganisms-11-00323-t005:** Methylation motifs and associated restriction–modification systems.

Strain Name	Methylation Motifs *	Identified Restriction–Modification Systems (RMSs) ^†^
*E. coli* 3/145	G**A***T*C	Dam M (425828..426664);
	**A**GCNNNNNC*T*TC	Type I EcoKI-like RMS (4380568..4384080);
*E. coli* 19/278	G**A***T*C	Dam M (395585..396421);
*K. pneumoniae* 13/97	G**A***T*C	Dam M (394809..395609);
*K. pneumoniae* 20/245	G**A***T*C	Dam M (383480..384307);
*P. aeruginosa* 7/157	C**A**GNNNNN*T*GGG	Type III M (1806087..1807313); Type I RMS (5234163..5240525); Type I SM (5245219..5248468)
*P. aeruginosa* 16/222	C**A**GNNNNN*T*GGG	Type III M (1810717..1811943); Type I RMS (5178388..5184750); Type I SM (5189444..5192693)
*P. aeruginosa* 9/195	G**A**CNNNNN*T*GCC	Type 1 RMS (5480535..5484830);
*S. pneumoniae* PHRX1-2021	G**A***T*C	Dam methylase (425828..426664)
	*T*CTAG**A**	Type II M XbaI (830119..831863);
	GA**A**NNNNNNNNN*T*TYG	Type I SMR (450315..455692); Type I SMR (467913..473290); Type I MR (481404..486290); Type I SMR (494068..498918); Type I MR (506310..510980); Type I MR (516392..520416); Type I SMR (1272436..1278431);
*S. epidermidis* 597	GGTG**A**	Type IIS MR HphI (513688..515338)
*S. aureus* 597/2	GW**A**GNNNNNN*T*AAA	Type I M (34076..35782); Type I R (179513..182302); Type I SM (411763..414571);
*S. aureus* 598	GW**A**GNNNNNGAT GG**A**NNNNNNN*T*CG	Type I R (164443..167232); Type I MS (402254..405035); Type I MS (1881251..1883954)

* Methylated nucleotides are shown in bold. In the case of a bipartite methylation at both DNA strands, the nucleotide opposing the methylation site on the reverse complement strand is italicized. **^†^** Restriction–modification systems (RMSs) may be composed of three subunits: M—methyltransferase; R—restriction endonuclease; S—motif recognition protein. The composition of the identified RMSs is indicated with the respective letters.

## Data Availability

Accession numbers of DNA sequences are given in Table 1.

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
