# Peer review of "Analysis of Whole-Genome Sequences of Pathogenic Gram-Positive and Gram-Negative Isolates from the Same Hospital Environment to Investigate Common Evolutionary Trends Associated with Horizontal Gene Exchange, Mutations and DNA Methylation Patterning"

_microorganisms, 2023, doi:10.3390/microorganisms11020323_

Round 1
Reviewer 1 Report
This study described characteristics of Gram positive and negative pathogens from a single hospital. This study is adequate, if describing these pathogens was the aim. However, the number of strains analyzed preclude from a comprehensive, population structure analysis even at a single hospital level. Future studies should employ a far higher number of strains to present a comprehensive analysis.
Abstract- Please add some data- species of strains, number of strains, etc.
Introduction: I would like to see some information about AMR in pathogens in Kazakhstan
Materials and methods: The description of strains is missing. Authors have mentioned previous studied, but this is too vital of information for authors to go searching for.
Results: This section is written rather blandly and should be summarized in a more impactful manner. At this point, it looks like I am reading through details of every single strain while missing a greater point.
Plasmids and virulence factor details should be added in a supplementary ( e.g. plasmid replicons, names of genes etc. can be uploaded as an excel sheet).
Contextualization of the metadata is missing from the article. e.g. if E coli were carrying genes that can cause UTI, were these E. coli then isolated from patients with UTI?
Discussion- It is short and adequate. Some of the discussion in results can be moved to discussion.
Author Response
This study described characteristics of Gram positive and negative pathogens from a single hospital. This study is adequate, if describing these pathogens was the aim. However, the number of strains analyzed preclude from a comprehensive, population structure analysis even at a single hospital level. Future studies should employ a far higher number of strains to present a comprehensive analysis.
Dear reviewer, thank you very much for your positive feedback to our paper. Indeed, the number of the strains used in this study is not sufficient for any reliable population statistics. The readers are explicitly warned about it in the Discussion, lines 682-687: “The number of strains used in this study is not sufficient to conclude which STs of the pathogens are the most abundant in the Syzganov’s National Scientific Center of Surgery in Almaty (Kazakhstan). However, even this small collection demonstrates that the bacterial population associated with the hospital environments may be quite versatile as the isolates of the same species belonged to different STs and were characterized with strain and ST specific repertoires of the factors of pathogenicity.”
The aim of the study was to consider possible commonalities and taxonomic/lineage specificities of virulence and antibiotic resistance development as this question, in contrast to the populational studies of different pathogens, is generally understudies. To highlight the specific aim of this paper, the following paragraphs were added:
Abstract, lines 19-21: “The aim of the study was to compare general and species-specific pathways of development of virulence and antibiotic resistance by opportunistic pathogens causing hospital acquired infections.”
Introduction, lines 86-93: “Numerous investigations were performed around the world to identify virulence factors and determine mechanisms of their acquisition by opportunistic pathogens causing sudden outbreaks of hospital infections. Based on these outputs, several useful databases of genetic virulence and antibiotic resistance determinants have been created [17-19]. However, the strategies of the majority of these studies were to use collections of bacterial strains belonging to the same species of a taxonomic group of pathogens isolated in an enclosed geographic region; whereas the co-evolutionary events in taxonomically distant pathogens inhabiting the same environment are generally understudied.”
Abstract- Please add some data- species of strains, number of strains, etc.
Thank you for your comment. This phrase was added in lines 16-19: “In this study, two Escherichia coli, two Klebsiella pneumoniae, three Pseudomonas aeruginosa, two Staphylococcus aureus, one Staphylococcus epidermidis and one Streptococcus pneumoniae showing broad spectra of antibiotic resistance were isolated from patients suffering from nosocomial infections in a local hospital in Almaty, Kazakhstan.”
Introduction: I would like to see some information about AMR in pathogens in Kazakhstan
Three recent reviews on the hospital acquired infections in Kazakhstan were discussed in the introduction (lines 75-85) and the respective references were added:
“Hospital acquired infections caused by various pathogens are frequent in hospitals of Central Asia, particularly in Kazakhstan. These infections are associated with surgical sites, followed by ventilator-associated pneumonia, catheter-related blood stream infections and catheter-associated urinary tract infections [14]. The most common causative agents of nosocomial infections in Kazakhstan are Pseudomonas aeruginosa, Escherichia coli, Klebsiella pneumoniae, Acinetobacter baumannii, Enterococcus faecalis, Staphylococcus epidermidis and S. aureus (MRSA and non-MRSA) [15, 16]. Many of these isolates showed broad spectra of resistance to methicillin, ceftriaxone, cefotaxime, cefuroxime and some other antibiotics; however, whole genome sequencing to determine specific genetic determinants of antibiotic resistance has not been performed in these studies.”
- Viderman, D., Khamzina, Y., Kaligozhin, Z., Khudaibergenova, M., Zhumadilov, A., Crape, B., Azizan, A. An observational case study of hospital associated infections in a critical care unit in Astana, Kazakhstan. Antimicrob. Resist. Infect. Control 2018, 7, 57. https://doi.org/10.1186/s13756-018-0350-0
- Viderman, D., Brotfain, E., Khamzina, Y., Kapanova, G., Zhumadilov, A., Poddighe, D. Bacterial resistance in the intensive care unit of developing countries: Report from a tertiary hospital in Kazakhstan. J. Glob. Antimicrob. Resist. 2019, 17, 35-38. https://doi.org/10.1016/j.jgar.2018.11.010
- Kaliyeva, S.S., Lavrinenko, A.V., Tishkambayev, Y., Zhussupova, G., Issabekova, A., Begesheva, D., Simokhina, N. Microbial Landscape and Antibiotic Susceptibility Dynamics of Skin and Soft Tissue Infections in Kazakhstan 2018-2020. Antibiotics (Basel) 2022, 11(5), 659. https://10.3390/antibiotics11050659
Materials and methods: The description of strains is missing. Authors have mentioned previous studied, but this is too vital of information for authors to go searching for.
Thank you for your comment. One additional column ‘Isolation metadata’ was added to Table 1 to describe shortly sources of isolation of these strains. Also, this table presents an information on antibiotic resistance patterns.
Also, a sentence was added in lines 110-111: “Routine confirmation of species identity was performed using NEFERMtest kit (Erba Lachema s.r.o., Czech Republic).” The phenotype of the strains, except for antibiotic resistance, was not very informative and won’t contribute to this paper. We are reluctant to add the results of biochemical tests to this paper as it will create an internal conflict of interests.
Results: This section is written rather blandly and should be summarized in a more impactful manner. At this point, it looks like I am reading through details of every single strain while missing a greater point.
Discussion- It is short and adequate. Some of the discussion in results can be moved to discussion.
Thank you for your comment. While the Result section was organized by the taxonomic groups of pathogens, in every paragraph a short summary was given on whether the observed virulence and antibiotic resistance determinants were common to other taxa or were species specific. But we agree with the reviewer that in the previous version this section contained some unnecessary details and too much discussions. Several paragraphs were removed and the paragraphs relevant to discussion were moved to the Discussion section as suggested.
Plasmids and virulence factor details should be added in a supplementary ( e.g. plasmid replicons, names of genes etc. can be uploaded as an excel sheet).
Thank you for your comment. The full list of the virulence genes located in the chromosomes and plasmids is given in the new Supplementary Table S2. This table is referenced in lines 452.
Contextualization of the metadata is missing from the article. e.g. if E coli were carrying genes that can cause UTI, were these E. coli then isolated from patients with UTI?
Thank you for your comment. One additional column ‘Isolation metadata’ was added to Table 1 to describe shortly the sources of isolation of these strains. The strain, E. coli 19/278 was isolated from urine from a patient suffering from pyelonephritis. The strain E. coli 3/145 was from wound discharge. This discovery was conceptualized in lines 478-484: “Discovery of NMEC virulence factors in the strains E. coli 3/145 and 19/278 is of importance as these genes encode the K1 capsular antigen that protects bacterial cells from phagocytosis. Other important virulence factors are the genes ibeBC supporting brain endothelial cell invasion; iss that protects pathogenic cells from phagocytosis; and traT that suppresses the immune response of the host. Interestingly, that all these factors are not common for the regular UPEC isolates [74]. It may indicate a trend towards generalizing the infection caused by these isolates”.
Reviewer 2 Report
Comments to the manuscript Microorganisms-2145944, entitled “Analysis of whole genome sequences of pathogenic Gram-positive and Gram-negative isolates from the same hospital environment to investigate common evolutionary trends associated with horizontal gene exchange, mutations and DNA methylation patterning”.
By analyzing the manuscript, it can be concluded that the authors did a great job; selected 11 multidrug-resistant strains from a large number of hospital isolates, sequenced their genomes using long-read PacBio sequencing, and performed a truly extensive data analysis. What further adds to the complexity is that the authors analyzed isolates belonging to three species of Gram-negative and two species of Gram-positive bacteria represented by several representatives each.
What is evident and to which I have no objections is the analysis of the presence of genes for antimicrobial resistance and virulence factors and their horizontal transfer within hospital isolates. The part dedicated to the mutational analysis is a bit controversial in my opinion, if the authors did not do Illumina sequencing in parallel. It is known that Illumina short reads sequencing is accurate, but cannot connect parts of the genome separated by repeated sequences, while long reads (PacBio and Nanopore) enable connection of separated parts but can introduce some changes. These errors are very rare, but they do exist and can give false results.
Accordingly, I believe that for accurate mutational analysis, hybrid assembly consisting of both reads should be performed.
The manuscript is quite well written with some minor flaws that I have indicated in the proposed suggestions/additions and changes.
The choice of references, I think could be a little better!
Conclusions could be more precise and of more general importance.
Minor suggestions:
Reference (Haque et al. 2018) is more adequate then [1].
Haque M, Sartelli M, McKimm J, Abu Bakar M. Health care-associated infections - an overview. Infect Drug Resist. 2018 Nov 15;11:2321-2333. doi: 10.2147/IDR.S177247.
Or WHO report on website:
https://www.who.int/news/item/06-05-2022-who-launches-first-ever-global-report-on-infection-prevention-and-control
Also, the reference (Peleg and Hooper, 2010) should be included in the citation related to the most common diseases acquired in hospitals.
Peleg AY, Hooper DC. Hospital-acquired infections due to gram-negative bacteria. N Engl J Med. 2010;362(19):1804-13. doi: 10.1056/NEJMra0904124. Together with [2-4].
Sikora A, Zahra F. Nosocomial Infections. [Updated 2022 Sep 23]. In: StatPearls [Internet]. Treasure Island (FL): StatPearls Publishing; 2022 Jan-. Available from: https://www.ncbi.nlm.nih.gov/books/NBK559312/
Beceiro A, Tomás M, Bou G. Antimicrobial resistance and virulence: a successful or deleterious association in the bacterial world? Clin Microbiol Rev. 2013;26(2):185-230. doi: 10.1128/CMR.00059-12. (together with [7])
Lines 44-45. Mutation of antibiotic target macromolecule is another mechanism of acquired drug resistance.
Not only proteins are targets for antibiotics (they can also be rRNAs, membranes, etc.)
Lines 90-92. Genomic DNAs were extracted from bacterial cells using PureLink Genomic DNA Kits (Publication Number: MAN0000601, Revision 2.0; Invitrogen) following the manufacturer’s instructions.
Lines 92-94. The integrity, size, quality and quantity of the isolated genomic DNA from bacterial strains were determined using the NanoDrop 2000c spectrophotometer (Thermo Scientific, USA) at the optical wavelengths of 260 nm and 280 nm and by electrophoresis on 1% agarose gels.
For long-read PacBio sequencing DNA fragments should be between 20-100 or even more kb and is recommended to be analysed on Pulsed Field Gel Electrophoresis.
Lines 151-154. Clinical isolates from Syzganov’s National Scientific Center of Surgery in Almaty (Kazakhstan) were tested for susceptibility to the following antibiotics:…….
I cannot get manuscripts [12-13] and consequently cannot understand many isolates were screened and on which media. Please could you put that data here; multidrug resistant isolates 11 of 700 were preselected for complete genome sequencing. In addition, Table with each species and antibiotic profile will be welcome.
Line 185. ……used for S. aureus MLST typing.
Line 202. 3.3. Plasmid and genomic island repertoire of the sequenced genomes
Lines 214-215. As one bacterial cell cannot contain several plasmids of the same incompatibility group, it…..
Why several, it could not be present two plasmids of the same incompatibility group in the same bacterial cell.
Line 218. … large plasmid from E. coli 3/145
It will be better as previously indicated plasmid 1 or plasmid 2 since both of them are large (or authors assume these two sequences as one large plasmid). All of this should be clearly explained.
Line 236. ……bacterial core genomes (Humphrey et al., 2021) [38].
Line 301. Mutations in EF-Tu and marR were found in all the E. coli and K. pneumoniae sequenced genomes.
Please when you are talking about strains put their names to be clear.
Author Response
By analyzing the manuscript, it can be concluded that the authors did a great job; selected 11 multidrug-resistant strains from a large number of hospital isolates, sequenced their genomes using long-read PacBio sequencing, and performed a truly extensive data analysis. What further adds to the complexity is that the authors analyzed isolates belonging to three species of Gram-negative and two species of Gram-positive bacteria represented by several representatives each.
Dear reviewer, thank you very much for your positive feedback.
What is evident and to which I have no objections is the analysis of the presence of genes for antimicrobial resistance and virulence factors and their horizontal transfer within hospital isolates. The part dedicated to the mutational analysis is a bit controversial in my opinion, if the authors did not do Illumina sequencing in parallel. It is known that Illumina short reads sequencing is accurate, but cannot connect parts of the genome separated by repeated sequences, while long reads (PacBio and Nanopore) enable connection of separated parts but can introduce some changes. These errors are very rare, but they do exist and can give false results.
Accordingly, I believe that for accurate mutational analysis, hybrid assembly consisting of both reads should be performed.
Thank you very much for your recommendation. Indeed, it was known from previous reports, publications and personal experience of many of us that the 3rd generation sequencers, i.e. SMRT PacBio and Oxford Nanopore, suffered from a lower base call quality. The fact that these technologies and the quality of the base calling have been improved dramatically in the last 5 years (CCS HiFi for PacBio and 2D sequencing for Nanopores) is missed by many researchers and many of us keep being suspicious regarding the quality of long reads generated by the 3rd generation sequencers. But it is not true anymore. More details were added to the section 2.2 regarding the DNA Library preparation and quality control. The following text was added in lines 132-136: “The DNA libraries were sequenced in Macrogen using the PacBio Sequel-I (Pacific Biosciences) sequencing platform exploiting the circular consensus sequencing (CCS) system producing HiFi (high fidelity) DNA reads ensuring 99.5% base call accuracy suitable for genome assembly and calling polymorphic sites [23]. The average coverage of reads per one nucleotide in different genomes was 150-200 [21].”.
We fully agree with the reviewer that the important mutations in individual genomes should be checked by re-sequencing to avoid false-positive predictions, but it is not possible when many strains are analysed and compared. The hybrid approach may be practical only in the case of a deep Illumina sequencing. Again, for a large number of strains it can be rather expensive. We decided to warn the readers that the reported mutations were not confirmed by resequencing and may be false-positives. I hope this statement in lines 373-376 will satisfy the reviewer: “The readers must be stressed that the polymorphic sites in the sequences were called by aligning long PacBio reads against the genome assembly with the aim to identify mutational commonalities between the different isolates. Individual mutations were not checked by re-sequencing and some level of false-positive predictions cannot be excluded”.
The manuscript is quite well written with some minor flaws that I have indicated in the proposed suggestions/additions and changes.
The choice of references, I think could be a little better!
Conclusions could be more precise and of more general importance.
Thank you very much for all these recommendations! Conclusion was modified. Also, we modified the discussion section by moving several paragraphs from the results to the discussion.
Minor suggestions:
Reference (Haque et al. 2018) is more adequate then [1].
Haque M, Sartelli M, McKimm J, Abu Bakar M. Health care-associated infections - an overview. Infect Drug Resist. 2018 Nov 15;11:2321-2333. doi: 10.2147/IDR.S177247.
- accepted, the reference was added
Or WHO report on website:
https://www.who.int/news/item/06-05-2022-who-launches-first-ever-global-report-on-infection-prevention-and-control
- the link was added in line 44
Also, the reference (Peleg and Hooper, 2010) should be included in the citation related to the most common diseases acquired in hospitals.
Peleg AY, Hooper DC. Hospital-acquired infections due to gram-negative bacteria. N Engl J Med. 2010;362(19):1804-13. doi: 10.1056/NEJMra0904124. Together with [2-4].
Sikora A, Zahra F. Nosocomial Infections. [Updated 2022 Sep 23]. In: StatPearls [Internet]. Treasure Island (FL): StatPearls Publishing; 2022 Jan-. Available from: https://www.ncbi.nlm.nih.gov/books/NBK559312/
- the reference and the text were added in lines 47-48.
Beceiro A, Tomás M, Bou G. Antimicrobial resistance and virulence: a successful or deleterious association in the bacterial world? Clin Microbiol Rev. 2013;26(2):185-230. doi: 10.1128/CMR.00059-12. (together with [7])
- the reference was added
Lines 44-45. Mutation of antibiotic target macromolecule is another mechanism of acquired drug resistance.
Not only proteins are targets for antibiotics (they can also be rRNAs, membranes, etc.)
- the phrase was modified in lines 51-52: “Mutation of the target macromolecules including proteins and functional RNAs is another mechanism of the acquired drug resistance”
Lines 90-92. Genomic DNAs were extracted from bacterial cells using PureLink Genomic DNA Kits (Publication Number: MAN0000601, Revision 2.0; Invitrogen) following the manufacturer’s instructions.
Lines 92-94. The integrity, size, quality and quantity of the isolated genomic DNA from bacterial strains were determined using the NanoDrop 2000c spectrophotometer (Thermo Scientific, USA) at the optical wavelengths of 260 nm and 280 nm and by electrophoresis on 1% agarose gels.
For long-read PacBio sequencing DNA fragments should be between 20-100 or even more kb and is recommended to be analysed on Pulsed Field Gel Electrophoresis.
- The major quality control and library preparation were done by the company Macrogene. Technical details provided by the company were added to the section 2.2: “the optical wavelengths of 260 nm and 280 nm and by electrophoresis on 1% agarose gels. The DNA library was prepared and quality controlled by Macrogen (Seoul, Korea). Library size and template size distribution were checked by running on a Agilent Technologies 2100 Bioanalyzer using a DNA 1000 chip. The library sample concentration was checked using Qubit standard Quantification solution.”
Lines 151-154. Clinical isolates from Syzganov’s National Scientific Center of Surgery in Almaty (Kazakhstan) were tested for susceptibility to the following antibiotics:…….
I cannot get manuscripts [12-13] and consequently cannot understand many isolates were screened and on which media. Please could you put that data here; multidrug resistant isolates 11 of 700 were preselected for complete genome sequencing. In addition, Table with each species and antibiotic profile will be welcome.
Two columns on antibiotic resistance profiles and isolation metadata were added to Table 1.
The text of the method section 1.1 was modifies, lines 117-120: “Zones of growth inhibition were measured after overnight cultivation of the bacterial cultures at 37ºC on solid CHROMagar Orientation medium ( bioMérieux, France) with Endo agar (for enterobacteria and staphylococci) or Cetrimide agar (for Pseudomonas).”
Line 185. ……used for S. aureus MLST typing.
- The text was changed: “They differ by sequences of 5 out of 7 aureus MLST marker genes.”
Line 202. 3.3. Plasmid and genomic island repertoire of the sequenced genomes
- The title was changed to “Plasmids and genomic islands of the sequenced isolates”
Lines 214-215. As one bacterial cell cannot contain several plasmids of the same incompatibility group, it…..
Why several, it could not be present two plasmids of the same incompatibility group in the same bacterial cell.
- Changed to two.
Line 218. … large plasmid from E. coli 3/145
It will be better as previously indicated plasmid 1 or plasmid 2 since both of them are large (or authors assume these two sequences as one large plasmid). All of this should be clearly explained.
- The number of the plasmid was added
Line 236. ……bacterial core genomes (Humphrey et al., 2021) [38].
- The error with this reference was fixed.
Line 301. Mutations in EF-Tu and marR were found in all the E. coli and K. pneumoniae sequenced genomes.
Please when you are talking about strains put their names to be clear.
- Everywhere the species names were added to the strain numbers. However, in the sentence above it is said that “all the E. coli and K. pneumoniae sequenced genomes”.
Once more, thank you very much for your useful comments and recommendations.
Authors of the manuscript.
Round 2
Reviewer 1 Report
Thank you for incorporating my suggestions.